# An exploratory open-label multicentre phase I/II trial evaluating the safety and efficacy of postnatal or prenatal and postnatal administration of allogeneic expanded fetal mesenchymal stem cells for the treatment of severe osteogenesis imperfecta in infants and fetuses: the BOOSTB4 trial protocol

Rachel L Sagar,[1,2] Eva Åström,[3,4] Lyn S Chitty,[5,6] Belinda Crowe,[7] Anna L David,[1,2] Catherine DeVile,[7] Annabelle Forsmark,[8] Vera Franzen,[9] Göran Hermeren,[10] Melissa Hill ![ORCID],[5,6] Mats Johansson,[10] Caroline Lindemans,[11] Peter Lindgren,[12,13] Wouter Nijhuis,[14] Dick Oepkes,[15] Mirko Rehberg,[16] Nils-Eric Sahlin,[10] Ralph Sakkers,[14] O Semler,[16] Mikael Sundin,[13,17] Lilian Walther-Jallow,[13] E J T Joanne Verweij,[15] Magnus Westgren,[13] Cecilia Götherström ![ORCID][13]

RLS and EÅ are joint first authors.

For numbered affiliations see end of article.

**Correspondence to**
Cecilia Götherström;
cecilia.gotherstrom@ki.se

## ABSTRACT

**Introduction** Severe osteogenesis imperfecta (OI) is a debilitating disease with no cure or sufficiently effective treatment. Mesenchymal stem cells (MSCs) have good safety profile, show promising effects and can form bone. The Boost Brittle Bones Before Birth (BOOSTB4) trial evaluates administration of allogeneic expanded human first trimester fetal liver MSCs (BOOST cells) for OI type 3 or severe type 4.

**Methods and analysis** BOOSTB4 is an exploratory, open-label, multiple dose, phase I/II clinical trial evaluating safety and efficacy of postnatal (n=15) or prenatal and postnatal (n=3, originally n=15) administration of BOOST cells for the treatment of severe OI compared with a combination of historical (1–5/subject) and untreated prospective controls (≤30). Infants<18 months of age (originally<12 months) and singleton pregnant women whose fetus has severe OI with confirmed glycine substitution in *COL1A1* or *COL1A2* can be included in the trial.

Each subject receives four intravenous doses of $3 \times 10^6$/kg BOOST cells at 4 month intervals, with 48 (doses 1–2) or 24 (doses 3–4) hours in-patient follow-up, primary follow-up at 6 and 12 months after the last dose and long-term follow-up yearly until 10 years after the first dose. Prenatal subjects receive the first dose via ultrasound-guided injection into the umbilical vein within the fetal liver (16+0 to 35+6 weeks), and three doses postnatally.

The primary outcome measures are safety and tolerability of repeated BOOST cell administration. The secondary outcome measures are number of fractures from baseline to primary and long-term follow-up, growth, change in bone mineral density, clinical OI status and biochemical bone turnover.

**Ethics and dissemination** The trial is approved by Competent Authorities in Sweden, the UK and the Netherlands (postnatal only). Results from the trial will be disseminated via CTIS, ClinicalTrials.gov and in scientific open-access scientific journals.

**Trial registration numbers** EudraCT 2015-003699-60, EUCT: 2023-504593-38-00, NCT03706482.

### STRENGTHS AND LIMITATIONS OF THIS STUDY

⇒ This is a protocol for an open-label clinical trial; which means that there is no randomisation and the treatment is known both to the families and the trial personnel.

⇒ Osteogenesis imperfecta is by nature a heterogeneous and variable disease, which may impact the possibility to satisfactorily evaluate the effect of treatments. To address this and collect reliable data, the Boost Brittle Bones Before Birth (BOOSTB4) trial collects data longitudinally and includes multiple follow-up assessments of each subject.

⇒ The trial may be underpowered to show improvements in efficacy. Therefore, if the trial yields a positive result, a larger subsequent multicentre pivotal trial may be needed.

## INTRODUCTION

Osteogenesis imperfecta (OI) is a rare clinically and genetically heterogeneous group

of heritable disorders of connective tissue characterised by osteopenia, growth restriction, bone deformities, fractures and chronic pain.[1] Originally, OI was classified into four types according to clinical, radiological and histological findings, where type 1 refers to the mildest form, type 2 to the perinatally lethal form, type 3 to the most severe form compatible with life and type 4 to an intermediate group with moderate to severe types of OI.[2] This classification covers 85%–90% of persons with OI and includes >1500 different autosomal dominant sequence variations in the genes coding for type I collagen (*COL1A1* or *COL1A2*).[1–3] There is currently no curative treatment for OI, with the only available pharmaceutical treatments failing to address the underlying molecular pathology, the collagen defect.[4] Treatment is aimed at increasing overall bone strength to prevent fractures and maintain mobility. Bisphosphonates are used to reduce bone resorption and increase bone mass.[5] However, two meta-analyses of randomised trials on bisphosphonate treatment for OI did not demonstrate improvement in fracture rates, reduction of pain or improved functional mobility, although multiple studies report this independently.[6 7]

One child among 10–20 000 is born with OI. Diagnosis of severe types of OI is often made at the routine fetal anomaly ultrasound scan mid-pregnancy, or earlier if first trimester scanning is routine. Early treatment of OI is preferred to minimise irreversible pathology. Prenatal intervention offers multiple potential advantages over postnatal approaches; ameliorating and preventing damage that has begun already during fetal life, the relatively naïve fetal immune system may permit the development of immune tolerance towards donor cells, infusion into the systemic fetal circulation may avoid sequestration of the cells in the lungs because circulatory shunts allow the pulmonary circulation to be bypassed, fetal life is a time of natural stem cell proliferation and migration to different anatomic compartments. Also, prenatal administration may enhance the effect of mesenchymal stem cells (MSCs) because there will be a lesser amount of poor quality OI type collagen that requires replacement, and finally, the possible psychological benefits for parents of commencing treatment as early in the child's life.[8]

The investigational medicinal product (IMP) investigated in the BOOSTB4 trial is an advanced therapy medicinal product consisting of allogeneic ex vivo expanded human fetal liver-derived MSCs from the first trimester of pregnancy. MSCs are multipotent stem cells and are promising candidates in regenerative medicine due to their low immunogenic profile that allows the possibility of performing cell transplantation across major histocompatibility barriers without immunosuppression.[9 10] In addition, they have the potential to differentiate along the osteogenic, chondrogenic and adipogenic lineages and have a good safety profile with minimal oncogenic risk.[11–16] Two meta-analyses of eight and 55 randomised controlled trials investigating intravascular administration of naïve MSCs to 321 and 2696 subjects, respectively, did not detect an association between non-fever acute

infusional toxicity, organ system complications, thrombotic/embolic events, infection, death or malignancy.[17 18] The only significant association with intravascular administration of MSCs was a transient fever.[17 18] The first clinical infusion of MSCs was performed over 20 years ago and adult MSCs have since been used in a diverse range of conditions in thousands of patients.[9 17] Studies show that MSCs derived from developmentally younger donor sources present characteristics suggesting a stronger cell therapeutic potential than adult MSCs, and this knowledge is being translated into clinical application. First trimester fetal MSCs are found at a higher frequency, are less mature on the transcriptomic level, have a greater colony-forming and proliferation capacity and are more immune modulatory compared with adult MSCs.[10 19–22] Interestingly, fetal MSCs differentiate into more lineages than MSCs derived from adult tissues, like skeletal muscle, oligodendrocytes and hepatic cells[23–25] and differentiate more readily into bone than adult MSCs. In a direct comparison of fetal MSCs and MSC derived from Wharton's Jelly, adult bone marrow and adipose tissue, fetal MSCs had higher levels of osteogenic genes under basal conditions.[26 27] On osteogenic differentiation, fetal MSCs produced more robust osteogenic gene expression, induced more calcium deposition in vitro and reached higher levels of osteogenic gene upregulation in vitro and in vivo.[26–28] First trimester fetal liver MSC were selected for use in this trial given their superior osteogenic differentiation capacity, their safety profile and the feasibility of manufacturing a high-quality Good Manufacturing Practice (GMP)-compliant off-the-shelf product with retained efficacy.

The potential of MSC transplantation for the treatment of OI has been shown in different mouse models of OI, with demonstrated homing and engraftment in bone, increased collagen content and mineralisation of the bones, improved bone strength, thickness and length and 67%–84% reduction in long bone fractures.[29–31] Clinically, two studies on intravenous administration of histocompatible MSCs from sibling donors to children with severe OI have been performed (six and two children treated, respectively, where six children underwent myeloablative haematopoietic stem cell transplantation before they received MSCs).[32–35] The MSC administration procedures were safe with no serious adverse events (AEs) reported, and positive clinical effects were attained with donor cell engraftment, development of new compact bone, improved bone parameters and 6 months following two MSC infusions, there was an increase in growth velocity, an 80% reduction in fracture frequency, an increase in total bone mineral content in 3/8 of the children[34 35] and in Quality of Life.[35 36]

We have previously reported two case studies of prenatal and postnatal transplantation of fetal MSCs for severe OI (type 3 and type 4 OI).[37 38] The limited data show that the procedures were safe and clinically promising when comparing the clinical course to other individuals with identical OI sequence variants. Only three

other individuals in the world are currently known to have a *COL1A2* variant identical to that of the patient with OI type 3; one of whom died from the severe type of OI despite bisphosphonate treatment and two individuals are described as having type 2/3 OI.[39 40] After one dose of fetal MSCs prenatally, donor cells were found in the recipient bones at 9 months of age, and the donor cells expressed bone markers.[37] Four booster doses with same donor MSCs between 8 and 13 years of age show a reduction in fracture rate (from 2/year to 0/year over 2 years), and an increase in height, although 5.0–6.4 SD below the mean.[38] While it is difficult to determine the definitive impact of the prenatal and postnatal MSC transplantation in these two heterogeneous cases, the findings suggest a potential clinical benefit. The children have now been followed up for over 21 and 15 years, respectively, and there has been no evidence of any early or late adverse reactions.

Donor cell engraftment in bones has been reported to be between 0% and 16.4% after stem cell transplantation in both human cases of OI and mouse models.[29–34 37 38] It is difficult to determine if such low levels of engraftment could result in adequate synthesis of healthy type I collagen and significant clinical improvement. However, Panaroni *et al* reported that in utero transplantation of healthy bone marrow to the Brtl OI mouse resulted in donor cell engraftment of 1%–2% in the bones, which in turn resulted in a 20% production of healthy collagen in the host bones.[30] The transplantation eliminated the perinatal lethality of Brtl mice, and furthermore the femora of treated Brtl mice had significant improvement in geometric parameters compared with Brtl mice and in mechanical properties compared with wild-type mice. Hence, low level engraftment of cells producing healthy collagen may lead to significant improvements in bone development.

Transplantation of MSCs is not a cure, but it is currently the only therapy available in clinical phase trials for OI that may result in healthy bone cells and production of healthy collagen. Other treatments developed for OI are based on modifying the action of the patient's own cells, and hence can only result in increased quantity of bone rather than improved quality of bone. However, the clinical effects of MSC administration may not be persistent and therefore redosing may be required. The interval for re-dosing is currently not known.

There is increasing interest in treatment with MSCs and a growing number of requests from patients and their physicians to provide it. In the absence of a clinical trial, prenatal and postnatal MSC transplantation for OI is likely to be practised around the world on a case-by-case basis, making it impossible to evaluate any effects that the intervention might have. Considering the complexity of this field and the distribution of cases, an international trial is the only realistic and ethically correct approach to enable accurate evaluation of the intervention. BOOSTB4 is the first clinical trial that will assess whether postnatal or prenatal and postnatal repeated intravenous

administration of MSCs is a safe and potentially effective treatment option for severe OI. The BOOSTB4 trial will also provide a well-defined framework for assessment of children and fetuses with OI treated with fetal MSCs.

## METHODS AND ANALYSIS
### Trial design
Boost Brittle Bones Before Birth (BOOSTB4) is an exploratory, open-label, multicentre, phase I/II trial evaluating safety and efficacy of four postnatal intravenous doses (n=15 infants) or one prenatal and three postnatal intravenous doses (n=3 fetuses in the amended protocol, in the original protocol it was n=15 fetuses) of allogeneic expanded fetal MSCs (BOOST cells) for the treatment of severe OI compared with a combination of historical (1–5/subject) and up to 30 untreated prospective controls. Please see the trial synopsis in online supplemental material.

### Setting
The Sponsor for this academic trial is Karolinska Institutet (coordinator) in Sweden. The trial sites are five maternity/fetal medicine and paediatric institutions; Karolinska University Hospital in Sweden (coordinating trial centre, OI and maternal/fetal medicine and manufacturer of the IMP), University College London (trial centre, maternal/fetal medicine) and Great Ormond Street Hospital for Children (trial centre, OI) in the UK and Leiden University Medical Centre (trial centre, maternal/fetal medicine) and University Medical Centre Utrecht (trial centre, OI) in the Netherlands.

### Investigational medicinal product
The IMP BOOST cells consists of cryopreserved expanded allogenic human first trimester liver-derived MSCs (the liver is the blood-forming organ at this time of development) obtained from early elective surgical terminations of pregnancy (see the Ethics and Dissemination section for information on donor information and consent). The IMP is manufactured under GMP, and the cells are isolated, expanded up to passage four and then cryopreserved as an off-the-shelf product ready to be thawed and reconstituted prior to administration. The cells are tested according to the Product Specification for sterility, endotoxin and mycoplasma with methods conforming to Pharmacopoeia, and they are further tested for expression of MSC markers by flow cytometry and for efficacy by a bone differentiation assay. The Mechanism of Action of the IMP is engraftment and bone differentiation. Two donors are used in the BOOSTB4 trial. At least 80 clinical doses of MSCs, and MSCs for retention samples and all safety and functionality analyses, will be manufactured from two donors.

### Eligibility and recruitment
The inclusion and exclusion criteria for the three trial groups are described in table 1. The upper age limit was

**Table 1** Inclusion and exclusion criteria for subjects in the BOOSTB4 trial

| Inclusion criteria | | |
|---|---|---|
| **Postnatal group (n=15)\*†** | **Prenatal group (n=3)\*‡§** | **Control group (n=18–120)\*** |
| 1. Parent's/legal guardian's signed informed-consent form<br>2. Clinical diagnosis of OI type 3 or severe 4 *AND*<br>3. Molecular diagnosis of OI (glycine substitution in the collagen triple-helix encoding region of either the *COL1A1* or *COL1A2* gene)<br>4. Age less than 18 months¶ (calculated from gestational week 40+0, ie, the corrected age)<br>5. Parent/legal guardian over 18 years of age | 1. Woman has signed the patient consent form<br>2. Only women where termination of the pregnancy is no longer possible or where the women are committed to continue the pregnancy may be included in the trial<br>3. Suspicion of OI type 3 or severe 4 in the fetus on ultrasound findings *AND*<br>4. Molecular diagnosis of OI in the fetus (glycine substitution in the collagen triple-helix encoding region of either the *COL1A1* or *COL1A2* gene)<br>5. Gestation age between 16+0 and 35+6 weeks+days<br>6. Pregnant women over 18 years of age | *Matched historical controls (1-5/subject):*<br>1. Parent's/legal guardian's signed informed-consent form<br>2. Clinical and molecular diagnosis of OI (glycine substitution in the collagen triple-helix encoding region of either the *COL1A1* or *COL1A2* gene)<br>3. Data on fractures and growth are available<br>4. Parent/legal guardian over 18 years of age<br>*Prospective untreated controls (n=up to 30):*<br>▶ Postnatal participation: the inclusion criteria for the postnatal group apply<br>▶ Prenatal participation: the inclusion criteria for the prenatal group apply, except inclusion criteria 2 |

| Exclusion criteria | | |
|---|---|---|
| **Postnatal group (n=15)** | **Prenatal group (n=3)§** | **Control group (n=18–120)** |
| 1. Existence of other known disorder that might interfere with the treatment, such as, but not limited to organ disfunction (eg, liver or renal failure or bronchopulmonary dysplasia), congenital heart defect, hypoxic encephalopathy I–III, severe neurological problems, immune deficiencies, muscle diseases, severe malformations or syndromes diagnosed by clinical examination<br>2. Any contraindication for invasive procedures such as a moderate/severe bleeding tendency<br>3. Known risk factors for clotting, such as, but not limited to previous blood clot, family history of clots, clotting disorder (inherited or acquired), heart failure, inflammatory disorders (eg, lupus, rheumatoid arthritis, inflammatory bowel disease)<br>4. Positive Donor Specific Antibody-test<br>5. Known allergy/hypersensitivity to Fungizone and/or Gensumycin<br>6. Abnormal karyotype or other confirmed genetic syndromes<br>7. Oncologic disease (previous or current malignancy)<br>8. Inability to comply with the trial protocol and follow-up schedule<br>9. Inability to understand the information and to provide informed consent | 1. Multiple pregnancy<br>2. Coexistence of other disorder that might interfere with the treatment, as judged by the Investigator or the patient's obstetrician<br>3. Abnormal fetal karyotype or other confirmed genetic syndrome<br>4. Any contraindication for invasive procedures such as a bleeding tendency or contagious infections, such as, but not limited to HIV, syphilis, hepatitis B, hepatitis C or other known infectious diseases that can harm the fetus<br>5. Known risk factors for clotting, such as, but not limited to previous blood clot, family history of clots, clotting disorder (inherited or acquired), heart failure, inflammatory disorders (eg, lupus, rheumatoid arthritis, inflammatory bowel disease)<br>6. Positive Donor Specific Antibody-test<br>7. Known allergy/hypersensitivity to Fungizone and/or Gensumycin<br>8. Oncologic disease in woman or fetus (previous or current malignancy)<br>9. Unwilling to or cannot undergo delivery by Caesarean section<br>10. Inability to comply with the trial protocol and follow-up schedule<br>11. Inability to understand the information and to provide informed consent | *Matched historical controls (1-5/subject):*<br>1. Existence of other disorder that might interfere with the trial. No lung hypoplasia (type 2 OI)<br>2. Abnormal karyotype<br>*Prospective untreated controls (n=up to 30):*<br>▶ Postnatal participation: the exclusion criteria, except exclusion criterium 2, 3, 4 and 5 (contraindication for invasive procedure, known risk factor for clotting, Positive Donor Specific Antibody-test and Known allergy/hypersensitivity to Fungizone and/or Gensumycin) for the postnatal group apply<br>▶ Prenatal participation: the exclusion criteria, except exclusion criterium 1, 4, 5, 6 and 7 (multiple pregnancy, contraindication for invasive procedure, known risk factor for clotting, Positive Donor Specific Antibody-test and Known allergy/hypersensitivity to Fungizone and/or Gensumycin) for the prenatal group apply |

For inclusion, all criteria should apply. For exclusion, one is enough to exclude.
Prenatal group: after birth and before the second dose, subjects in the prenatal group will be assessed for the inclusion and exclusion criteria for the postnatal group.
\*All subjects (the postnatal, prenatal and control groups) will receive bisphosphonate treatment.
†At least one dose of bisphosphonate must be administered before the first dose of stem cells.
‡The first dose of bisphosphonates will be administered after birth.
§In the original trial protocol, the prenatal group was n=15. This amendment has been approved by the Competent Authorities.
¶In the original trial protocol, it was 'Age less than 12 months'. This amendment has been approved by the Competent Authorities.
BOOSTB4, Boost Brittle Bones Before Birth; OI, osteogenesis imperfecta.

amended from maximum 12 months of age to 18 months of age in the postnatal group. This change to the original protocol has been approved by the Competent Authorities, and was implemented to allow for inclusion of more subjects to the postnatal group. All subjects (the postnatal, prenatal and control groups) will receive bisphosphonate treatment. At least one dose of bisphosphonate must be administered before the first dose of stem cells in the postnatal group. In the prenatal group, the first dose of bisphosphonates will be administered after birth.

Acceptable types of concomitant medication including bisphosphates during the trial and until the 12 month follow-up are pamidronate (preferred) or neridronate, and the annual dose of bisphosphate is 8–12 mg/kg body weight per year.

Information about the BOOSTB4 trial has been distributed to specialist OI services and fetal medicine units in Europe, and the BOOSTB4 trial team can also be contacted directly by families or physicians via email or the trial specific website (www.boostb4.eu) where the Patient Information Leaflet is available in several languages. The potential subjects are invited to visit the trial site or have an organised video meeting and provide their oral and written consent to be screened for participation. At least 3 days will be required for the parents/legal guardians to consider participation in the trial. For all children enrolled in the trial, a signed informed-consent form for the main trial is required from the mother and the father, if she/he has parental responsibility (see online supplemental material for the screening and main trial participation consent forms for the postnatal and prenatal groups, and www.boostb4.eu for the Participant information sheets). For prenatal inclusion, only the pregnant woman needs to sign the informed-consent form, but both parents are encouraged to sign. After the birth of a child in the prenatal group, the inclusion and exclusion criteria for the postnatal group must be fulfilled (table 1), and both parents must reconsent their participation in the trial before the first postnatal dose. Consent may be withdrawn at any time without a reason being stated. The Sponsor will hold clinical trial insurance that covers all enrolled subjects receiving active treatment.

## Sample size calculation

No formal power calculations has been performed. OI is a rare disease, and a limited number of subjects can be included each year. Based on the incidence, it was originally reasonable to include 30 patients (15 postnatal subjects and 15 prenatal subjects) in this first trial on the grounds of feasibility, suitable time period and for the application of descriptive statistical analyses. However, in an amendment to the original protocol in 2023, which is approved by the Competent Authorities, the number of subjects in the prenatal group was reduced from 15 to three. This change was requested by the Sponsor due to time restrictions in the academic funding of the trial. The risk–benefit balance for participation in any of the trial groups was not changed. Since the purpose of the prenatal trial group was to demonstrate feasibility of administration of one dose of BOOST cells before birth and three doses after birth, this change to the original protocol is not considered to alter the outcome of the trial's objectives.

## Blinding, intervention and follow-up

It would not be ethically appropriate to perform a randomised double-blind placebo-control trial in this population, particularly regarding the risks of the prenatal

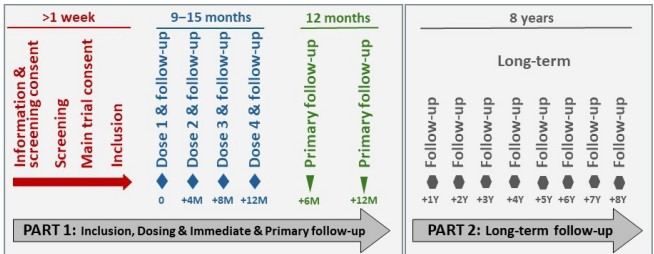

**Figure 1** Overview of the trial schedule. The BOOSTB4 trial consists of two parts; part 1 that includes information, consent, screening and inclusion. This is followed by four doses of BOOST cells at 4±1 month intervals with immediate in-patient follow-up over 48 hours after doses 1–2, and over 24 hours after doses 3–4 (48 hours for doses 1–3 in the prenatal group). Part 1 of the trial also includes the primary follow-up at 6 and 12 months after the fourth and last dose. Part 2 of the trial consists of yearly long-term follow-up over 8 years until 10 years after the first dose. BOOST, Boost Brittle Bones Before Birth.

infusion procedure and the vulnerability of the subjects. No blinding or randomisation will therefore take place.

Four intravenous doses of BOOST cells are administered at 4 month intervals (range 3–5 months), please see figure 1 for an overview of the trial. For postnatal subjects, the first dose must be administered before 18 months of age (in the original protocol, it was before 12 months of age; an amendment that has been approved by the Competent Authorities). For prenatal subjects, the first dose will be given in utero during gestational weeks+days 16+0–35+6 via ultrasound-guided injection into the umbilical vein within the fetal liver, and the final three doses are administered postnatally. Each dose contains $3 \times 10^6$ allogeneic expanded human fetal liver MSCs per kg bodyweight. All doses to a subject will be derived from one donor. No preconditioning or medicines after the administration will be administered.

In-patient follow-up is performed over 48 (doses 1–2) or 24 (doses 3–4) hours in the postnatal group and over 48 (doses 1–3) or 24 (dose 4) hours in the prenatal group after each dose. Primary follow-up is performed at 6 and 12 months after the last dose, and long-term follow-up is performed yearly until 10 years after the first dose. At each visit and follow-up, AEs and parameters related to efficacy outcomes are recorded. Women in the prenatal group are followed under the same schedule.

Prospective controls will be subject to the same follow-up routine as the patients enrolled in the treatment group, as described above, with the option to opt in or out from blood collection. The clinical evaluation programme for the historical control group will be based on data available from the routine follow-up of OI patients in the respective country. The controls will be individually matched to each subject (see table 2). Between one and five historical controls will be included per subject and up to 30 prospective controls in total.

**Table 2** Matching of the controls

| Matching of the controls | |
|---|---|
| The controls historical and untreated prospective will be individually matched to each trial subject: | |
| Type of OI*: | ► Type 3 with type 3.<br>► Severe type 4 with severe type 4. |
| Age at onset of bisphosphonate therapy: | ► Start of bisphosphonate treatment in the first 2 months of life.<br>► Start of bisphosphonate treatment from 2 to 12 months of life. |
| If possible (if more than five controls have been identified), matching will also be made on: | |
| Sex | |
| Identical OI pathogenic sequence variant | |
| Time for OI diagnosis | ► A control matched with a subject in the postnatal group was diagnosed†<br>postnatally.<br>► A control matched with a subject in the prenatal group was diagnosed†<br>prenatally. |

One to five historical controls per subject and up to 30 prospective controls in total will be included and matched to each subject.
*No mixed OI types (eg, OI type 3/4, is allowed).
†Clinical diagnosis, molecular diagnosis not required at this time point.
OI, osteogenesis imperfecta.

## Outcome measures

The primary endpoint of the BOOSTB4 trial is safety and tolerability measured as seriousness, severity and frequency of administration-related AEs in the child, fetus and woman after four postnatal or one prenatal and three postnatal intravenous doses of BOOST cells every 4 months. Specific focus will be given to vital signs, transfusion and immune reactions and maternal and fetal AEs using the Maternal and Fetal AE Terminology,[41] for example, persistent fetal bradycardia following the infusion, tumorigenicity, morbidity and mortality. The trial has clearly defined Stopping criteria on trial, group and subject levels. In case of a serious adverse reaction, the subject may not receive more doses of BOOST cells. The Data and Safety Monitoring Board will review all AEs and adverse reactions.

Vital signs (oxygen saturation, heart rate, respiratory rate, body temperature, blood pressure (only in the pregnant woman), medical examination and signs of suspected pulmonary embolism) are monitored in the child and pregnant woman before, during and until 48 hours (doses 1–2) or 24 hours (doses 3–4) after each dose. In the fetus, the following is monitored before, during and until 48 hours after the prenatal dose using ultrasound: fetal heart rate, middle cerebral artery pulsatility index, peak systolic velocity, umbilical artery Doppler pulsatility index and cardiotocography.

The following parameters are analysed in peripheral blood before, and at 1, 24 (doses 1–4) and 48 hours (doses 1–2) after each dose (does not apply to the fetus): blood status (haemoglobin, haematocrit, leukocytes, thrombocytes, erythrocytes and mean corpuscular volume), liver (ALAT and ASAT), kidney (Creatinine) and electrolytes (Chloride, Magnesium, Sodium and Potassium). Donor specific antibodies are analysed before the first dose and >3 weeks after each dose.

The secondary endpoints relate to efficacy of four doses of BOOST cells in children with OI. The hypothesis under investigation is that early administration of BOOST cells will ameliorate severe types of OI. The key secondary endpoint is number of fractures from baseline to primary (12 months after the last dose) and long-term (10 years after the first dose) follow-up. Fracture frequency was chosen as the key secondary endpoint because it is the most clinically relevant efficacy variable. Additional secondary endpoints are time to first fracture after each BOOST cell administration, number of fractures at birth in the prenatal group, growth (length/height and weight), change in bone mineral density measured with DXA/DEXA (not a trial specific investigation, optional and according to the national follow-up programme), clinical OI status and biochemical bone turnover in peripheral blood. The analysis of biochemical bone turnover includes Calcium, Phosphate, Albumin, Parathyroid hormone, 25-OH Vitamin D, Alkaline phosphatase, Bone-specific Alkaline phosphatase, Osteocalcin and cross-linked C-telopeptide of type I collagen, and will be carried out before each dose and then according to the clinical routine every 6–12 months. No radiological measurements will be performed specifically for the trial.

The exploratory endpoints investigated by the trial include assessing the impact of the therapy on Quality of Life using the Infant Toddler Quality of Life Questionnaire (ITQOL), collection of peripheral blood before and at 1 and 24 hours (doses 1–4) and 48 hours (doses 1–2) after each dose for studying the paracrine effect (via the Olink technology or ELISA) and the effect on endogenous immune cells (via flow cytometry) of BOOST cells in subjects and prospective controls (blood sampling optional). The extent of donor cell engraftment in tissue samples collected from routine surgery will be studied, and a method for non-invasive method for prenatal

diagnosis of OI developed (not used as the diagnostic method in the trial). In addition, there will be a substudy to BOOSTB4, where parents will have the option to take part in two semistructured interviews, in which their experiences of their participation in the trial are explored. This will allow us to understand their reasoning for their child to participate in the trial, and what has been positive and what can be improved in future similar studies.

## Statistical analysis

The primary endpoint of the BOOSTB4 trial is the safety of postnatal or prenatal and postnatal administration of BOOST cells. It involves investigation of the seriousness, severity and frequency of treatment-related AEs in the child, woman and fetus. All reported AEs (regardless of the cause) will be included in the statistical analysis, and all analyses will be performed separately for the infant/child, fetus and pregnant woman. No imputation of missing data will take place.

The BOOSTB4 trial is a non-randomised trial and specific caution will be taken to control for problems with bias. The statistical analysis will include descriptive statistics of the data that will be discussed in relation to the general knowledge of the condition, to generate an overall picture of the safety and efficacy profile of the treatment. All trial subjects will be compared with himself or herself as his or her own control over time (longitudinally from baseline to the primary follow-up at 12 months after the last dose and to the long-term follow-up until 10 years after the first dose). In the analysis of parameters such as blood status and biochemical bone turnover, non-parametric Wilcoxon matched pairs signed-rank test (before vs after doses and over time) will be used. One to five matched historical controls (18–90 in total) will, as far as possible, be included per trial subject to increase the statistical precision, and up to 30 prospective untreated controls will also be included.

## Patient and public involvement

The trial has been discussed with patient support groups including the OI Federation Europe (OIFE) and Care 4 Brittle Bones (C4BB) from early in the planning stage. We are grateful for their input, and for the joint dissemination of the trial in 2019.

In a separate study, we explored stakeholder views to understand perceived benefits or concerns, identify ethical issues and establish protocols for support and counselling. Semistructured qualitative interviews were conducted with 56 participants from three groups: adults affected with OI, with and without children and parents of children affected with OI, health professionals who work with patients with OI and patient advocates from relevant patient support groups.[8] There were generally positive views towards using fetal MSC transplantation to treat OI, with early treatment considered to be advantageous for preventing fractures and reducing severity, as well as potentially bringing psychological benefits for parents. Participants were somewhat concerned about

procedure safety, short and long-term side effects and the effectiveness of transplantation. The difficulties inherent in decision-making were frequently raised, as well as the importance of explaining fetal MSC transplantation in a way that all parents can understand, setting clear expectations, providing psychological support and allowing time to reflect during the decision-making process. The feedback received has been incorporated into the design and planning of the BOOSTB4 trial.

## Health economic assessment

A health economic assessment is being undertaken with the objective to describe the costs and effects of BOOST cell administration as a potential standard treatment for severe forms of OI. Published data on the impact of OI on healthcare burden are scarce,[42] with available studies reporting a high cost for, for example, frequent hospital admissions.[43 44] Also, two sets of questionnaires have been developed to capture dependency on healthcare and/or community services, with one focusing on patient/family (postnatal and prenatal groups) and the other on maternal health (prenatal group). These questionnaires will be used to further assess the costs and effects of BOOST cell transplantation for severe forms of OI, in addition to further exploring the economic burden of OI overall.

## Trial status

Recruitment commenced in Sweden after the site opened on 12 August 2019. The first subject in the postnatal group was included and received the first dose on 17 March 2020. Due to the SARS-CoV-2 pandemic, only two subjects were included in the trial during 2020, and due to time restrictions for the project, the dose administrations are only performed at Karolinska University Hospital in Sweden. After a safety assessment by the Data and Safety Monitoring Board of at least one dose administered to the first five postnatal subjects in the trial, the prenatal group of the trial was opened on 17 June 2021, and the first prenatal dose was administered on 18 November 2021. By 2nd of March 2022, all 15 of 15 subjects in the postnatal group had been included and received their first dose, and by 14th of December 2021, three of 15 prenatal subjects had been included and administered with one prenatal dose each. No more subjects will be included in the prenatal group due to time restrictions in funding of the trial (an amendment approved by the Competent Authorities). The primary follow-up is currently being performed and is expected to be completed in January 2024, and the long-term follow-up will be completed in 2032.

## ETHICS AND DISSEMINATION

The BOOSTB4 clinical trial aims to investigate BOOST cell treatment in young children and unborn children (which includes the pregnant woman as a research subject) suffering from a chronic disease. The trial has obtained

ethical and regulatory approval from the national Competent Authorities in Sweden (the Swedish Ethical Review Authority (Etikprövningmyndigheten), approval no 2018/1732-31 and the Swedish Medical Product Agency, approval no 5.1-2018-57411), UK (the North East—York Research Ethics committee, approval no 19/NE/0030 and the UK Medicines and Healthcare products Regulatory Agency, approval no 44119/0002/001-001), Netherlands (the Central Committee on Research Involving Human Subjects (Centrala Commissie Mensgebonden Onderzoek), approval no NL64105.000-17, postnatal part). The NHS Research Ethics Committee and the North of Scotland Research Ethics Committee separately approved the interview study in the UK (approval no 16/NS/0084), which were part of the approved trial protocol in Sweden and the Netherlands. Amendments have been implemented after approval from the National Competent Authorities. An approved amendment included reducing the prenatal trial group from 15 subjects to three subjects (the Swedish Ethical Review Authority, approval no 2023-03725-02 and the Swedish Medical Product Agency, approval no 5.1-2023-38290). Oral and written informed consent will be obtained from the legal guardians to all subjects/the pregnant women (for prenatal inclusion), please refer to the section eligibility and recruitment for details. The trial is performed according to Good Clinical Practice. The data are entered into electronic Case Report Forms, and the data and trial are monitored according to the trial's Monitoring Plan. The conduct of the trial is overseen by an independent Data and Safety Monitoring Board according to their Charter. The Data and Safety Monitoring Board includes expertise on OI, haematology, transplantation and fetal and maternal medicine.

Several ethical issues arise when considering stem cell transplantation for OI in these study populations. Therefore, a detailed and systematic risk–benefit assessment was performed before initiation of the trial, which is updated when needed, for example during the COVID pandemic. The assessment considers many areas; risks related to handling, administration and exposure to the IMP, and risks for the child, fetus and pregnant woman. The assessment also includes potential benefits for the child, fetus and the parents. All risks and benefits are discussed in relation to the Sponsor's experience with administration of fetal MSCs for treatment of OI. In other words, this therapy is believed to be directly beneficial to the fetus and child, but there are also significant unknowns.

In the BOOSTB4 trial, there have been several separate considerations of diverse ethical issues. They include reviewing the ethical and legal standards relevant to prenatal and postnatal stem cell transplantation, mapping the rights and interests of the research subjects, investigating stakeholder's views,[8] as well as testing the Patient Information Leaflet and the communication of risks.

The fetal MSCs used for manufacturing the IMP BOOST cells are obtained from first trimester fetal tissues after elective termination of pregnancy. Ethical approval has been obtained in Sweden for the use of fetal tissues for manufacture of the IMP (the Swedish Ethical Review Authority, approval no 2018/1732-31 and the National Board of Health and Welfare, approval no 8.1-28985/2018). The donating woman independently seeks medical care. Oral and written information about the study is given by a medical practitioner not related to the study, who also obtains informed oral and written consent. A health declaration and infectious screening is performed. The woman can withdraw from the study until the IMP has been manufactured.

Fetal tissues might pose an issue with regard to ethical as well as religious beliefs for some individuals, and transparency in this is vital. To address this, we performed interviews with 56 participants as explained above under patient and public involvement. The interviews revealed generally positive views towards using fetal MSCs for treating severe OI.[8]

Identifying the main stakeholders is fundamental since the safety and rights of the research subjects should be the main concern. At the prenatal stage, the main stakeholders are the pregnant woman and the fetus. At the postnatal stage, only the minor is the research subject, but the parents are also stakeholders, since the infant cannot consent to the participation. We have also discussed the risk of overprotection of pregnant women since they are considered to be a vulnerable population. This may violate their autonomy, harm their interests and prevent research that could help children and possibly improve their own Quality of Life. Overprotection may signal good intentions and an ambition to minimise risks, but it could also result from a tendency to underestimate the benefits the therapy may bring to the women from whom consent is being sought. Partner consent is not legally required prenatally, although there are exceptions (discussed in ref.[45]). Nonetheless, it is recommended if the partner is expected to become the child's legal representative after birth. It will also be impossible to complete the trial if the partner dissents after birth, and the trial concerns the partner in many ways.

As a substudy to BOOSTB4, the parents can consent to take part in two qualitative interviews. This will allow us to understand their reasoning for their child to participate in the trial, and what has been positive and what can be improved in future similar studies.

During 2024, the trial protocol will be transferred to the European Union's Clinical Trials Information System (CTIS) (previously EudraCT). Until then, the information on EudraCT is not updated. The EUCT number for the trial is 2023-504593-38-00. Please refer to ClinicalTrials.gov for the most recent information. The trial results will be uploaded on CTIS and ClinicalTrials.gov and published in scientific open-access journals when the primary follow-up has been completed, and also when and the long-term follow-up has been completed. Lay summaries of the findings will be shared via the networks of the OI patient organisations OIFE and C4BB, and distributed to the families in the BOOSTB4 trial that have opted to

receive such summaries. Details of the trial and progress will be shared by presentation at relevant national and international meetings.

**Author affiliations**
[1]Elizabeth Garrett Anderson Institute for Women's Health, University College London, London, UK
[2]NIHR University College London Hospitals Biomedical Research Centre, London, UK
[3]Department of Women's and Children's Health, Karolinska Institutet, Stockholm, Sweden
[4]Astrid Lindgren Children's Hospital, Karolinska University Hospital, Stockholm, Sweden
[5]North Thames Genomic Laboratory Hub, Great Ormond Street Hospital for Children NHS Foundation Trust, London, UK
[6]Genetics and Genomics, UCL Great Ormond Street Institute of Child Health, London, UK
[7]Department of Neurosciences, Great Ormond Street Hospital for Children NHS Foundation Trust, London, UK
[8]PharmaLex, Gothenburg, Sweden
[9]XNK Therapeutics AB, Huddinge, Sweden
[10]Department of Clinical Sciences, Lund University Faculty of Medicine, Lund, Sweden
[11]Department of Pediatrics, University Medical Centre Utrecht, Utrecht, The Netherlands
[12]Center for Fetal Medicine, Karolinska University Hospital, Stockholm, Sweden
[13]Department of Clinical Science, Intervention and Technology, Karolinska Institutet, Stockholm, Sweden
[14]Department of Orthopedic Surgery, University Medical Centre Utrecht, Utrecht, The Netherlands
[15]Department of Obstetrics, Leiden University Medical Center, Leiden, The Netherlands
[16]Department of Pediatrics, University Hospital Cologne, Koln, Nordrhein-Westfalen, Germany
[17]Section of Pediatic Hematology, Immunology and HCT, Karolinska University Hospital, Stockholm, Sweden

**Acknowledgements** We thank all families taking part in the BOOSTB4 trial, and physicians collaborating in accomplishing the trial. We also thank OIFE and C4BB for their constructive and valuable support.

**Contributors** CG, MW, EÅ, OS and ALD conceived the idea for the trial. All authors (RSL, EÅ, LSC, BC, ALD, CDV, AF, VF, GH, MH, MJ, CL, PL, WN, DO, MR, N-ES, RS, OS, MS, LW-J, EJTJV, MW and CG) developed the trial protocol. CG, MW, EÅ and LW-J wrote the initial trial protocol, which was subsequently revised by all authors. GH, MJ and N-ES wrote the initial texts on ethics and regulatory issues. EÅ developed the clinical practice of osteogenesis imperfecta follow-up. MS developed the clinical practice and clinical safety parameters, and developed the study database. MH and LSC designed the research with stakeholders, MH conducted the research with stakeholders and contributed to the development of parent facing study materials. RS, EÅ and CG wrote the draft of the manuscript and all authors (RLS, EÅ, LSC, BC, ALD, CDV, AF, VF, GH, MH, MJ, CL, PL, WN, DO, MR, N-ES, RS, OS, MS, LW-J, EJTJV, MW and CG) revised, reviewed and approved the final manuscript.

**Funding** This project has received funding from the European Union's Horizon 2020 research and innovation programme under grant agreement No 681045, Vetenskapsrådet (The Swedish Research Council, 921-2014-7209 and 2020-01666), Region Stockholm in Sweden (20200365, FoUI-975495), the Center for Innovative Medicine (CIMED) in Sweden (FoUI-962010) and the Foundation for Children of the Freemasons in Stockholm. Parts of this work were supported by grants from the German Science Foundation (Deutsche Forschungsgemeinschaft, DFG) to the Research Unit FOR 2722; Refnr SE2373/1-2, Project 384170921. LSC is partially funded by the NIHR Great Ormond Street Hospital (GOSH) Biomedical Research Centre (BRC). All research at Great Ormond Street Hospital NHS Foundation Trust and UCL Great Ormond Street Institute of Child Health is made possible by the NIHR Great Ormond Street Hospital. ALD is partially funded by the NIHR University College London Hospitals BRC.

**Disclaimer** The views expressed are those of the authors and not necessarily those of the NHS, the NIHR or the UK Department of Health.

**Competing interests** ALD is a consultant for Esperare Foundation for a clinical trial unrelated to this work. CG, LW-J and MW are cofounders and coowners of BOOST Pharma ApS founded in 2020. OS is a scientific advisor for BOOST Pharma ApS.

**Patient and public involvement** Patients and/or the public were involved in the design, or conduct, or reporting or dissemination plans of this research. Refer to the Methods section for further details.

**Patient consent for publication** Not applicable.

**Provenance and peer review** Not commissioned; externally peer reviewed.

**ORCID iDs**
Melissa Hill http://orcid.org/0000-0003-3900-1425
Cecilia Götherström http://orcid.org/0000-0003-3782-0048

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
