## [Reviewer comments · BMJ Open]

ARTICLE DETAILS

TITLE (PROVISIONAL)	An exploratory open label multicentre phase I/II trial evaluating the safety and efficacy of postnatal or pre- and postnatal administration of allogeneic expanded fetal mesenchymal stem cells for the treatment of severe Osteogenesis Imperfecta in infants and fetuses: The BOOSTB4 Trial Protocol
AUTHORS	Sagar, Rachel; Åström, Eva; Chitty, Lyn; Crowe, Belinda; David, Anna; DeVile, Catherine; Forsmark, Annabelle; Franzen, Vera; Hermeren, Göran; Hill, Melissa; johansson, mats; Lindemans, Caroline; Lindgren, Peter; Nijhuis, Wouter; Oepkes, Dick; Rehberg, Mirko; Sahlin, Nils-Eric; Sakkers, Ralph; Semler, O.; Sundin, Mikael; Walther-Jallow, Lilian; Verweij, E.J.T; Westgren, Magnus; Götherström, Cecilia

VERSION 1 – REVIEW

REVIEWER	Rodríguez, Clara Cruces University Hospital, Biobizkaia Health Research Instit
REVIEW RETURNED	17-Oct-2023

GENERAL COMMENTS	The trial protocol presented is adequate and accurate for the ambitious proposed objective of evaluating the safety and efficacy of prenatal and pre-and postnatal administration of fetal mesenchymal stem cells for the treatment of severe pediatric rare disorder Osteogenesis Imperfecta. There are some suggestions that would add more clarity to the trial protocol and the trial itself that should be considered: .-Information regarding the MSCs isolation and characterization should be added. How many patients will be treated with the BOOSTB4 cells coming from a single donor?.-“Patients undergoing bisphosphonate treatment regardless of when they started it” should be added to the inclusion criteria..-The clinical evaluation will be performed at 12 months after the last MSCs infusion and 10 years. That means there are only 3 monitoring visits; before treatment, at 12 months and at 10 years after the last MSCs infusion? When will peripheral blood be collected for biochemical analyses of bone turnover? At the same time as the 3 monitoring visits?.-Will be the control group subjected to a similar monitoring visits (clinical evaluation and peripheral blood collection) in those period of time?.-Regarding the secondary endpoints related to the efficacy of cell treatments, in addition to the biochemical analysis of bone turnover, it would be interesting to also compare the molecular composition of sera/plasma collected before vs the plasma/sera
---

	collected after the cell treatment in each patient. Those results should be compared with similar analysis in the control group in case the sera/plasma samples at the 3 monitoring visits were available.
--	--

REVIEWER	Bishop, Nick University of Sheffield
-----------------	---

REVIEW RETURNED	27-Oct-2023
-------------

GENERAL COMMENTS	Thank you for asking me to review this protocol. The authors have not reported any results, but have completed recruitment to the post-natal part of the protocol that administers fetal MSCs. I have a few comments in relation to the background information provided and specific comments in relation to the assessment of efficacy and patient involvement. Background The authors state that MSCs, either fetally derived or from matched sibling donors, have been previously shown to have specific effects on bone outcomes in a total of ten children. Six of these children were previously enrolled in a study of the effects of bone marrow transplantation; the authors state (using ref 25 as evidence) that in these six there were significant improvements in bone mineral content (BMC) following administration of MSCs. However, careful reading of ref 25 reveals that only one of the six patients studied showed any increase in BMC. The other cited reports relate to two patients enrolled in the TERCELOI study (refs 26, 27) and one treated by the Karolinska team (29) and one treated in Singapore (28) reported with further data from the Karolinska patient. Reference 27 is cited as showing MSCs in bone but this should be ref 28, I think. It is clear that the MSC infusions had a good safety profile in all the studies. The efficacy reporting is much more difficult to assess, as the baselines for the individual cases were not reported according to a standardised methodology, and all the children continued to receive bisphosphonates during the period when the MSCs were administered. Despite the findings of persistent expression of donor-derived genes within bone tissue post-MSCTransplantation, it is difficult to declare with certainty that the level of normal type I collagen synthesis, processing, export and incorporation into the extracellular matrix will be sufficient to alter the clinical phenotype in more severely affected individuals where the on-going production of abnormal type I collagen will continue to exert a dominant-negative effect. Methodology/efficacy One particular issue that may be difficult for the authors to address within the current protocol design is the well-known variability in severity of OI within families that have the same genetic mutation; there are likely to be environmental as well as genetically-determined factors that influence the clinical course in any given affected individual and a robust baseline assessment of disease severity is needed to convince the reader that a significant improvement is due to the procedure of MSC infusion. I cannot see in the protocol whether other bone-active treatments such
---

	as bisphosphonates are an exclusion. If patients are able to continue to receive such additional treatment it is going to be very difficult to determine the true efficacy of MSC treatment. How do the authors propose to address this issue? Of note, regulators have insisted on placebo-controlled RCTs for other OI-directed interventions. The authors state in the abstract that the study is likely too small to provide evidence of efficacy, and then state that “if the trial yields a positive result, a larger subsequent multi-centre pivotal trial may be needed.” We already know, based on the information provided by the authors, that the primary outcome of “safety” is likely to be met. It would be helpful for the authors to define what “a positive result” actually means, and exactly when that determination would be made – it is currently stated that the assessments would be at 12 months after last dose and ten years after first dose – I wonder if the latter is a realistic follow-up period, especially if age-matched contemporaneous control data are required. Patient involvement It is great to see the strong involvement of patients as well as professionals in the design of the study. Is there on-going involvement of patients in the trial management processes, and an expectation of their involvement in the dissemination of the study results?
--	---

REVIEWER	Naserian, Sina INSERM, Inserm U1197
REVIEW RETURNED	02-Nov-2023

GENERAL COMMENTS	Paper number: bjmopen-2023-079767 Paper title: An exploratory phase I/II trial evaluating the safety and efficacy of postnatal or pre-and postnatal administration of allogeneic expanded fetal mesenchymal stem cells for the treatment of severe Osteogenesis Imperfecta: The BOOSTB4 Trial Protocol Authors: Rachel L Sagar, Eva Åström, Lyn S Chitty, Belinda Crowe, Anna L David, Catherine DeVile, Annabelle Forsmark, Vera Franzén, Göran Hermerén, Melissa Hill, Mats Johansson, Caroline A Lindemans, Peter Lindgren, Wouter H Nijhuis , Dick Oepkes, Mirko Rehberg, Nils-Eric Sahlin, Ralph Sakkers, Oliver Semler, Mikael Sundin, Lilian Walther-Jallow, Joanne Verweij, Magnus Westgren and Cecilia Götherström. General comments: This work focuses on severe osteogenesis imperfecta (also known as Brittle Bones disease), a disease for which there is currently no effective treatment. In this protocol paper, the authors focus on a phase I/II trial in which the authors will evaluate the administration of allogeneic expanded human first-trimester fetal liver mesenchymal stem cells (MSCs) for osteogenesis imperfecta type III or severe type IV (which are subtypes of Brittle Bones Diseases). This is the first clinical trial that assess whether postnatal or pre-and postnatal administration of allogeneic expanded human fetal liver-MSC cells are a safe and effective treatment option for OI type III or severe type IV.
--

	The first objective of the trial is to assess safety and tolerability in the fetus, child, and woman after postnatal or prenatal and postnatal intravenous administration of four doses of fetal MSC cells in individuals with OI type III or severe type IV. The trial is also aimed at increasing bone strength to prevent fractures and maintain mobility. Evaluation is performed before and after all dose administration and is then performed at 6 and 12 months after the last dose. Thereafter the subjects are followed yearly until 10 years after the first dose. The team behind this work has been working on fetal-liver MSC properties effects over the past 20 years during which they have demonstrated much innovative information, particularly on fetal MSC and their potential application in regenerative medicine. The phase I/II trial described in this paper is well-designed, scientifically credible, and reported according to the appropriate guidelines of competent authorities. Generally, the authors should emphasize fetal versus adult MSC properties, in order to show more clearly the incremental advance of fetal use over adult MSC use in such trials. In general, many references concerning fetal MSC properties are missing in the introduction and the discussion sections and should be added to complement the text. The article may only become acceptable after minor revisions of content as per the reviewer's comments. Specific comments: The abstract page 3 is too short and therefore not sufficiently informative. In the "Methods and Analysis" paragraph lines 39-40 page 3 and lines 46-47 page 10 the authors wrote, "Prenatal subjects receive the first dose via ultrasound-guided intrahepatic umbilical vein injection". The authors should change the sentence that the injection cannot be "intrahepatic intra umbilical vein". In the "Ethics and dissemination" paragraph page 3 the authors should give the name of the competent authorities in Sweden, the United Kingdom and Netherland. The authors should also indicate the specificity of each partner in the clinical trial. It is well known that the clinical effect after MSC transplantation is not permanent, and regular injections are needed. It may be that prenatal or prenatal and postnatal serial transplantations may not be clinically sufficient for permanent bone function amelioration. Could the authors add this point in the "Strengths and limitations of this study" paragraph, page 4, as a limitation to this clinical trial? In the introduction chapter the authors should develop the paragraph on MSC differentiation properties The authors wrote "In addition, they have the potential to differentiate along many lineages...". Could the authors describe which lineages exactly? The authors should also mention the MSC immunosuppressive properties in addition to their immunologic properties. With the same aim of improving understanding of MSC safety and tolerability, the authors should develop paragraphs devoted to these aspects in the introduction section.
--	--

	The authors should add more references in the introduction chapter focusing on the characteristics of MSC depending on the tissue types they originated from (e.g. PMID: 31239604). In the introduction section the authors wrote” First trimester fetal MSCs are found at a higher frequency, are less mature and have a greater colony-forming and proliferation capacity compared to adult MSCs. Could the authors explain on which criteria they claim that fetal MSCs are less mature than adult MSC and develop this aspect in the introduction section? When the authors address the differences observed between MSC derived from fetal and adult tissues the authors only mention work on MSC osteogenesis capacity MSC. The authors should develop this part, and extend the introduction to other properties of fetal cells that seem as important as osteogenic capacities in such a clinical trial. The authors should at least mention previous published work on fetal MSC immunologic and immunosuppressive properties versus adult MSC. Indeed, the prenatal transplantation approach is well justify as the immunological status of the fetus may ensure the development of immune tolerance towards the donor cells rendering postnatal transplantations more efficient. The authors should also justify the use of fetal MSC versus adult MSC in link with their immunological profile and to their immunosuppressive properties in addition to others properties (frequency, “maturity” and proliferative capacity) highlighted in the paper by the authors. Concerning the MSC osteogenic capacity the author wrote « Interestingly, fetal MSCs differentiate more readily into bone than adult MSCs, and in a direct comparison of fetal MSCs and MSC derived from Wharton’s Jelly, adult bone marrow and adipose tissue, fetal MSCs had higher levels of osteogenic genes under basal conditions. Upon osteogenic differentiation, fetal MSCs produced more robust osteogenic gene expression, induced more calcium deposition in vitro and reached higher levels of osteogenic gene upregulation in vitro and in vivo ». In the cited studies fetal MSC are not from fetal liver origin but from fetal bone marrow origin (PMID: 18832592 and PMID: 19836073). Moreover a hierarchy was observed within fetal samples, with fetal bone marrow MSC having greater osteogenic potential than fetal blood MSC, which in turn had greater osteogenic potential than fetal liver MSC (PMID: 18557767). Could the author discuss that point that seem crucial to understand while the authors chose fetal liver MSC and not fetal bone marrow MSC for their clinical trial ? In the “Outcome Measures” paragraph the authors wrote “Additional secondary endpoints are time to first fracture after each BOOST cell administration, number of fractures at birth in the prenatal group, growth (length/height and weight), change in bone mineral density measured with DXA/DEXA, clinical OI status and biochemical bone turnover in peripheral blood ». Could the authors describe which parameters are used to evaluate the biochemical bone turnover in peripheral blood? In the “Statistical Analysis” chapter could the authors precise which type of statistical method (hypothesis test) in addition to the descriptive statistics will be used and which statistical parameters will be used?
--	---

	A graphical summary is missing and a schematic representation of the experimental protocol should be provided, which would help to understand the experimental procedure and the molecular mechanisms involved in severe osteogenesis imperfecta.
--	---

VERSION 1 – AUTHOR RESPONSE

Reviewer: 1

Dr. Clara Rodríguez , Cruces University Hospital

Comments to the Author:

The trial protocol presented is adequate and accurate for the ambitious proposed objective of evaluating the safety and efficacy of prenatal and pre-and postnatal administration of fetal mesenchymal stem cells for the treatment of severe pediatric rare disorder Osteogenesis Imperfecta.

There are some suggestions that would add more clarity to the trial protocol and the trial itself that should be considered:

-Information regarding the MSCs isolation and characterization should be added. How many patients will be treated with the BOOSTB4 cells coming from a single donor?

RESPONSE: Information about the IMP has been added to page 10 in the manuscript:

“The IMP consists of cryopreserved expanded human first trimester liver-derived MSCs (BOOST cells), obtained from early elective surgical terminations of pregnancy. The Mechanism of Action of the IMP is engraftment and bone differentiation. The IMP is manufactured under Good Manufacturing Practice (GMP) and is tested with regards to safety and functionality according to the approved product specification. At least 80 clinical doses of MSCs, and MSCs for retention samples and all analyses will be manufactured from two different donors used throughout the BOOSTB4 trial.”.

Unfortunately, an in-depth description of the manufacturing and associated testing is beyond the scope and space limitations of the current manuscript.

-“Patients undergoing bisphosphonate treatment regardless of when they started it” should be added to the inclusion criteria.

RESPONSE: All participants (active participation and historical and prospective controls) in the trial will receive bisphosphonates according to their country’s national treatment program. All participants in the postnatal group must have received at least one dose of bisphosphonates before the first dose of stem cells. The prenatal participants will receive the first dose of bisphosphonates after birth. The following text has been added to page 10 (and Table 1) of the manuscript:

“All participants (the postnatal, prenatal and control groups) will receive bisphosphonate treatment. At least one dose of bisphosphonate must be administered before the first dose of stem cells in the postnatal group. In the prenatal group the first dose of bisphosphonates will be administered after birth. Acceptable types of concomitant medication including bisphosphates during the trial and until the 12-month follow-up are Pamidronate (preferred) or Neridronate, and the annual dose of bisphosphate is 8-12 mg/kg body weight per year.”

-The clinical evaluation will be performed at 12 months after the last MSCs infusion and 10 years. That means there are only 3 monitoring visits; before treatment, at 12 months and at 10 years after the last MSCs infusion? When will peripheral blood be collected for biochemical analyses of bone turnover? At the same time as the 3 monitoring visits?

RESPONSE: The planned follow-up is described on page 12: “In-patient follow-up is performed over 48 (dose 1–2) or 24 (dose 3–4) hours in the postnatal group and over 48 (dose 1–3) or 24 (dose 4) hours in the prenatal group after each dose. Primary follow-up is performed at 6 and 12 months after the last dose, and long-term follow up is performed yearly until 10 years after the first dose. At each

visit and follow-up, AEs and parameters related to efficacy outcomes are recorded. Women in the prenatal group are followed under the same schedule.”.

Hence, follow-up is planned to be performed at 14 occasions over 10 years, and of course in between planned follow-up if required. For clarity and so no faulty association is made to monitoring of the conduct of the trial, we have changed “in-patient monitoring” to “in-patient follow-up” in the abstract and on page 12, and we have also added an overview of the trial and the follow-up visits as Figure 1. Regarding sampling for biochemical bone turnover, the following text has been added on page 13: “The analysis of biochemical bone turnover includes Calcium, Phosphate, Albumin, Parathyroid hormone, 25-OH Vitamin D, Alkaline phosphatase, Bone specific Alkaline phosphatase, Osteocalcin and cross-linked C-telopeptide of type I collagen, and will be carried out before each dose and then according to the clinical routine every 6–12 months.”

-Will be the control group subjected to a similar monitoring visits (clinical evaluation and peripheral blood collection) in those period of time?

RESPONSE: Data from the historical controls will be collected from medical files after informed consent. Hence, the historical controls will not be subjected to any trial visits. The prospective controls will be asked to fill out trial questionnaires and data will be collected at the same time points as in the trial. According to consent, blood may be collected from prospective controls.

The following text can be found on page 12: “Prospective controls will be subject to the same follow-up routine as the patients enrolled in the treatment group, as described above, with the option to opt in or out from blood collection. The clinical evaluation program for the historical control group will be based on data available from the routine follow-up of OI patients in the respective country.”

-Regarding the secondary endpoints related to the efficacy of cell treatments, in addition to the biochemical analysis of bone turnover, it would be interesting to also compare the molecular composition of sera/plasma collected before vs the plasma/sera collected after the cell treatment in each patient. Those results should be compared with similar analysis in the control group in case the sera/plasma samples at the 3 monitoring visits were available.

RESPONSE: We are indeed analysing the paracrine effects using plasma/sera collected before and after the dose administrations, which has been clarified on page 13.

Reviewer: 2

Nick Bishop, University of Sheffield

Comments to the Author:

Thank you for asking me to review this protocol. The authors have not reported any results, but have completed recruitment to the post-natal part of the protocol that administers fetal MSCs. I have a few comments in relation to the background information provided and specific comments in relation to the assessment of efficacy and patient involvement.

Background

The authors state that MSCs, either fetally derived or from matched sibling donors, have been previously shown to have specific effects on bone outcomes in a total of ten children. Six of these children were previously enrolled in a study of the effects of bone marrow transplantation; the authors state (using ref 25 as evidence) that in these six there were significant improvements in bone mineral content (BMC) following administration of MSCs. However, careful reading of ref 25 reveals that only one of the six patients studied showed any increase in BMC. The other cited reports relate to two patients enrolled in the TERCELOI study (refs 26, 27) and one treated by the Karolinska team (29) and one treated in Singapore (28) reported with further data from the Karolinska patient. Reference 27 is cited as showing MSCs in bone but this should be ref 28, I think. It is clear that the MSC infusions had a good safety profile in all the studies. The efficacy reporting is much more difficult to assess, as the baselines for the individual cases were not reported according to a standardised

methodology, and all the children continued to receive bisphosphonates during the period when the MSCs were administered. Despite the findings of persistent expression of donor-derived genes within bone tissue post-MSC transplantation, it is difficult to declare with certainty that the level of normal type I collagen synthesis, processing, export and incorporation into the extracellular matrix will be sufficient to alter the clinical phenotype in more severely affected individuals where the on-going production of abnormal type I collagen will continue to exert a dominant-negative effect.

RESPONSE: We have added information that 6 of the children in the study by Horwitz were treated with myeloablative haematopoietic stem cell transplantation before they received MSCs and added two references related to this study (reference 32 and 33) on page 7. Also, we have specified that an increase in bone mineral content was demonstrated in 3/8 children in the studies by Horwitz et al. and Infante et al. (page 7).

We agree with the reviewer that assessing efficacy of treatments is complicated in OI since the field lacks standardized methods. Therefore, including multiple parameters in the assessment, as done in the BOOSTB4 trial, may result in a more reliable conclusion. Off label bisphosphonates are today standard of care and are typically administered to all patients with severe types of OI. With this concomitant medication of bisphosphonates in the clinical trials, it is imperative that any comparisons are made to controls also receiving bisphosphonates on a similar treatment program. In BOOSTB4, all controls receive bisphosphonates (please refer to page 13 of the manuscript).

Panaroni et al. reported that in utero transplantation of healthy bone marrow to the Brtl OI mouse resulted in donor cell engraftment of 1–2% in the bones, which in turn resulted in a 20% production of healthy collagen in the host bones (Panaroni, Gioia et al. 2009). The transplantation eliminated the perinatal lethality of Brtl mice, and furthermore the femora of treated Brtl mice had significant improvement in geometric parameters compared to Brtl mice and mechanical properties compared to wild-type mice. Hence, low level engraftment of cells producing healthy collagen may lead to significant improvements. It is important to remember that administration of stem cells will not cure OI, but it is the only treatment in clinical phase for OI that may result in healthy bone cells and production of healthy collagen. Other treatments currently developed for OI are based on altering the turnover of the patient's own cells in the bones, and hence they will only result in more bone of poor quality. This has been added to the "Strengths and limitations of this study" part in the manuscript.

Methodology/efficacy

One particular issue that may be difficult for the authors to address within the current protocol design is the well-known variability in severity of OI within families that have the same genetic mutation; there are likely to be environmental as well as genetically-determined factors that influence the clinical course in any given affected individual and a robust baseline assessment of disease severity is needed to convince the reader that a significant improvement is due to the procedure of MSC infusion. I cannot see in the protocol whether other bone-active treatments such as bisphosphonates are an exclusion. If patients are able to continue to receive such additional treatment it is going to be very difficult to determine the true efficacy of MSC treatment. How do the authors propose to address this issue? Of note, regulators have insisted on placebo-controlled RCTs for other OI-directed interventions. The authors state in the abstract that the study is likely too small to provide evidence of efficacy, and then state that "if the trial yields a positive result, a larger subsequent multi-centre pivotal trial may be needed." We already know, based on the information provided by the authors, that the primary outcome of "safety" is likely to be met. It would be helpful for the authors to define what "a positive result" actually means, and exactly when that determination would be made – it is currently stated that the assessments would be at 12 months after last dose and ten years after first dose – I wonder if the latter is a realistic follow-up period, especially if age-matched contemporaneous control data are required.

RESPONSE: The variability in severity of OI within families and persons that have the same pathogenic variant is indeed difficult to address and overcome, which is a problem for all trials on OI. The field have started to address this issue by performing "natural history" studies, but it will take a

long time before they have been completed, and moreover, they are not focused on the pediatric population. We believe it is still possible to perform trials on young children with OI, especially since they are the population that will benefit most from finding an efficient treatment. We have addressed this issue e.g. by collecting data at baseline from birth to the first dose, and then longitudinally over a long period of time to follow each child's progress. Controls (1–5 historical controls/subject, and up to 30 prospective controls) are matched to each subject using the trial's matching criteria (added on page 13 and as Table 2). The pathogenic sequence variant is matching criteria number 4 of 5, and hence the foremost determining matching criteria. We have added the complexity of performing a clinical trial on OI to the "Strengths and Limitations of this study" section of the manuscript.

In the BOOSTB4 trial all subjects (active and controls) are receiving the standard of care, which includes bisphosphonates. We have clarified this on page 12 and in Table 1. All historical controls will have received bisphosphonates since this has been the standard of care for many years. If the trial subjects did not receive bisphosphonates, the trial would thus not have any relevant controls. It is not ethically acceptable at this early stage, to perform a randomized placebo-controlled trial on this vulnerable and protected population consisting of children with severe types of OI and healthy pregnant women. Additionally, the benefit-risk balance of stem cell administration to children with milder OI types is not favorable at this early developmental stage. The trial set-up was agreed upon together with the National Competent Authorities in Europe, Sweden, the Netherlands and the United Kingdom, and the regulators did not insist on a placebo-controlled randomized controlled trial.

Initial case studies performed on administering MSCs to patients with OI speak for the safety of the treatment, but no systematic clinical trial using an off-the-shelf product has been performed until now, highlighting the importance to perform such regulated clinical trials to determine the safety of the treatment and administration procedures. The BOOSTB4 trial is firstly a safety trial, and the trial protocol states that the "Definition of a successful trial" is fulfilment of the primary objective safety: Definition of a successful trial

Fulfilment of the primary objective safety in the postnatal group is defined as absence of procedure and/or treatment-related:

1. Mortality,
2. Tumourigenicity,
3. Transmissible disease,
4. Immune and/or administration reactions with severe and/or persistent effects on the health,
5. Severe effects on vital signs in conjunction to the administration,
6. Other morbidities with severe long-term effect on the health.

Fulfilment of the primary objective safety in the prenatal group (woman and fetus/child) is defined as absence of procedure and/or treatment-related:

1. Mortality,
2. Tumourigenicity,
3. Transmissible disease,
4. Immune and/or administration reactions with severe and/or persistent effects on the health,
5. Severe effects on vital signs in conjunction to the administration,
6. Adverse effects of feto-maternal transmission of donor cells,
7. Other morbidities with severe long-term effect on the health.

Regarding follow-up time, please see the RESPONSE to comment 3 by reviewer 1. All subjects will be followed at each of the 4 doses, at 6 and 12 months after the last dose, and then yearly for 8 years (please also see Figure 1 that has been added to explain the trial overview). The longer follow-up time is common in trials on cell- and gene therapies. Baseline data from birth to the first dose is collected. The same program is followed for the controls.

Patient involvement

It is great to see the strong involvement of patients as well as professionals in the design of the study. Is there on-going involvement of patients in the trial management processes, and an expectation of their involvement in the dissemination of the study results?

RESPONSE: We greatly value the contributions made by the patient organizations and the multidisciplinary array of professionals; their input has been and still is of vast importance and has improved BOOSTB4. However, patients/parents to the patients are not actively involved in the trial management process of BOOSTB4, and are not expected to. We are in regular contact with the Osteogenesis Imperfecta Federation Europe (OIFE) who have expressed an interest in the trial and wish to be informed about the progress of the trial. We plan to collaboratively disseminate the results of the trial together with OIFE.

Reviewer: 3

Dr. Sina Naserian, INSERM

Comments to the Author:

General comments:

This work focuses on severe osteogenesis imperfecta (also known as Brittle Bones disease), a disease for which there is currently no effective treatment.

In this protocol paper, the authors focus on a phase I/II trial in which the authors will evaluate the administration of allogeneic expanded human first-trimester fetal liver mesenchymal stem cells (MSCs) for osteogenesis imperfecta type III or severe type IV (which are subtypes of Brittle Bones Diseases).

This is the first clinical trial that assess whether postnatal or pre- and postnatal administration of allogenic expanded human fetal liver-MSC cells are a safe and effective treatment option for OI type III or severe type IV.

The first objective of the trial is to assess safety and tolerability in the fetus, child, and woman after postnatal or prenatal and postnatal intravenous administration of four doses of fetal MSC cells in individuals with OI type III or severe type IV. The trial is also aimed at increasing bone strength to prevent fractures and maintain mobility.

Evaluation is performed before and after all dose administration and is then performed at 6 and 12 months after the last dose. Thereafter the subjects are followed yearly until 10 years after the first dose.

The team behind this work has been working on fetal-liver MSC properties effects over the past 20 years during which they have demonstrated much innovative information, particularly on fetal MSC and their potential application in regenerative medicine.

The phase I/II trial described in this paper is well-designed, scientifically credible, and reported according to the appropriate guidelines of competent authorities.

Generally, the authors should emphasize fetal versus adult MSC properties, in order to show more clearly the incremental advance of fetal use over adult MSC use in such trials. In general, many references concerning fetal MSC properties are missing in the introduction and the discussion sections and should be added to complement the text.

RESPONSE:

We thank the reviewer for addressing this. We have added additional relevant references in the manuscript addressing this, for e.g. the study by Yu et al. on page 6, reference 21 (Yu, Valderrama et al. 2021).

The article may only become acceptable after minor revisions of content as per the reviewer's comments.

Specific comments:

The abstract page 3 is too short and therefore not sufficiently informative.

RESPONSE: Currently the abstract includes 300 words, and only 300 words are allowed. The abstract must include specific information such as Introduction; Methods and analysis; Ethics and dissemination, and Registration details.

In the "Methods and Analysis" paragraph lines 39-40 page 3 and lines 46-47 page 10 the authors wrote, "Prenatal subjects receive the first dose via ultrasound-guided intrahepatic umbilical vein injection". The authors should change the sentence that the injection cannot be "intrahepatic intra umbilical vein".

RESPONSE: Thank you for the comment. We have clarified this procedure and have changed the wording from "...via ultrasound-guided intrahepatic umbilical vein injection" to "...via ultrasound-guided injection into the umbilical vein within the fetal liver" (abstract and page 11).

In the "Ethics and dissemination" paragraph page 3 the authors should give the name of the competent authorities in Sweden, the United Kingdom and Netherland. The authors should also indicate the specificity of each partner in the clinical trial.

RESPONSE: This information has been added to "Setting" on page 9.

It is well known that the clinical effect after MSC transplantation is not permanent, and regular injections are needed. It may be that prenatal or prenatal and postnatal serial transplantations may not be clinically sufficient for permanent bone function amelioration. Could the authors add this point in the "Strengths and limitations of this study" paragraph, page 4, as a limitation to this clinical trial?

RESPONSE: This was a good suggestion, and we have added a bullet point on page 4 stating the following: "...the clinical effects of MSC administration may not be persistent and therefore re-dosing may be required. The interval for re-dosing is currently not known."

In the introduction chapter the authors should develop the paragraph on MSC differentiation properties. The authors wrote "In addition, they have the potential to differentiate along many lineages...". Could the authors describe which lineages exactly? The authors should also mention the MSC immunosuppressive properties in addition to their immunologic properties. With the same aim of improving understanding of MSC safety and tolerability, the authors should develop paragraphs devoted to these aspects in the introduction section.

RESPONSE: More information about the differentiation capacity and the immunologic properties of fetal MSCs have been added to the introduction. We have also added more information on the known safety and tolerability of MSCs to the introduction.

The authors should add more references in the introduction chapter focusing on the characteristics of MSC depending on the tissue types they originated from (e.g. PMID: 31239604).

RESPONSE: We agree with the reviewer that more knowledge and understanding is needed about MSCs derived from different tissues and developmental stages. Unfortunately, the scope of this manuscript describing the BOOSTB4 clinical trial protocol does not permit a detailed elaboration on these points. The rationale for using fetal liver as the starting material for manufacture of the IMP investigated in the BOOSTB4 trial is described in the introduction, and additional information about the rationale of using fetal liver as the starting material has been added to page 7.

In the introduction section the authors wrote "First trimester fetal MSCs are found at a higher frequency, are less mature and have a greater colony-forming and proliferation capacity compared to

adult MSCs. Could the authors explain on which criteria they claim that fetal MSCs are less mature than adult MSC and develop this aspect in the introduction section?

RESPONSE: The fetal MSCs are less mature when their transcriptome has been compared to MSCs from adult bone marrow (Götherström, West et al. 2005, Guillot, Gotherstrom et al. 2007). This information has been added to the introduction.

When the authors address the differences observed between MSC derived from fetal and adult tissues the authors only mention work on MSC osteogenesis capacity MSC. The authors should develop this part, and extend the introduction to other properties of fetal cells that seem as important as osteogenic capacities in such a clinical trial. The authors should at least mention previous published work on fetal MSC immunologic and immunosuppressive properties versus adult MSC. Indeed, the prenatal transplantation approach is well justify as the immunological status of the fetus may ensure the development of immune tolerance towards the donor cells rendering postnatal transplantations more efficient. The authors should also justify the use of fetal MSC versus adult MSC in link with their immunological profile and to their immunosuppressive properties in addition to others properties (frequency, "maturity" and proliferative capacity) highlighted in the paper by the authors.

RESPONSE: As described above, the text has been updated to include more information and additional references have been added, but the lack of space does not permit detailed descriptions. Since the aim of the BOOSTB4 clinical trial is to treat the bone disease Osteogenesis Imperfecta, we chose to focus on properties by the fetal MSCs that are related to this. The Mechanism of Action of the IMP is engraftment and bone differentiation (added to page 10). In the trial we primarily investigate safety and tolerability of the fetal MSCs, and secondly efficacy is investigated, which includes endpoints related to the Mechanism of Action such as fracture frequency, growth and bone mineral density. The exploratory endpoints include paracrine effects and effects on immune cells. The introduction is therefore focused on these parameters in relation to the characteristics of fetal MSCs.

Concerning the MSC osteogenic capacity the author wrote « Interestingly, fetal MSCs differentiate more readily into bone than adult MSCs, and in a direct comparison of fetal MSCs and MSC derived from Wharton's Jelly, adult bone marrow and adipose tissue, fetal MSCs had higher levels of osteogenic genes under basal conditions. Upon osteogenic differentiation, fetal MSCs produced more robust osteogenic gene expression, induced more calcium deposition in vitro and reached higher levels of osteogenic gene upregulation in vitro and in vivo ». In the cited studies fetal MSC are not from fetal liver origin but from fetal bone marrow origin (PMID: 18832592 and PMID: 19836073). Moreover a hierarchy was observed within fetal samples, with fetal bone marrow MSC having greater osteogenic potential than fetal blood MSC, which in turn had greater osteogenic potential than fetal liver MSC (PMID: 18557767). Could the author discuss that point that seem crucial to understand while the authors chose fetal liver MSC and not fetal bone marrow MSC for their clinical trial ?

RESPONSE: We are aware of the described hierarchical difference in the cited study. During the first trimester the MSCs migrate from the yolk sac, to the fetal liver and lastly to the bone marrow later during development. Although the fetal bone marrow and blood show a more osteogenic profile in one study, it is our honest belief that fetal bone marrow and blood are not viable sources for manufacture of a GMP compliant off-the-shelf product. For the manufacture of a cell therapy product many aspects must be considered such as number of cells in the starting material versus the need to expand the cells to obtain enough cells for the treatment, possible contaminating cells, methods for procuring the tissues, to mention a few. Of note is that all MSCs derived from fetal first trimester tissues (liver, bone marrow and blood) are reported to possess a higher osteogenic differentiation capacity compared to MSCs from other sources or developmental stages. We have added the rationale for using fetal liver as the starting material to page 7.

In the "Outcome Measures" paragraph the authors wrote "Additional secondary endpoints are time to first fracture after each BOOST cell administration, number of fractures at birth in the prenatal group, growth (length/height and weight), change in bone mineral density measured with DXA/DEXA, clinical

OI status and biochemical bone turnover in peripheral blood ». Could the authors describe which parameters are used to evaluate the biochemical bone turnover in peripheral blood?

RESPONSE: The analysis of the biochemical bone turnover in peripheral blood includes Calcium, Phosphate, Albumin, Parathyroid hormone, 25-OH Vitamin D, Alkaline phosphatase, Bone specific Alkaline phosphatase, Osteocalcin and cross-linked C-telopeptide of type I collagen. This information has been added to page 13.

In the "Statistical Analysis" chapter could the authors precise which type of statistical method (hypothesis test) in addition to the descriptive statistics will be used and which statistical parameters will be used?

RESPONSE: In agreement with the National Competent Authorities, only descriptive statistics in relation to the general knowledge of the condition and the clinical progress of each child will be used. As described, all trial subjects will be compared to himself or herself as his or her own control over time (longitudinally from baseline to the primary follow-up at 12 months after the last dose and to the long-term follow-up until 10 years after the first dose). In the analysis of parameters such as vital signs, blood status and biochemical bone turnover, non-parametric Wilcoxon matched pairs signed-rank test will be used (before versus after doses and over time). The statistics section in the manuscript has been updated to include the latter part.

A graphical summary is missing and a schematic representation of the experimental protocol should be provided, which would help to understand the experimental procedure and the molecular mechanisms involved in severe osteogenesis imperfecta.

RESPONSE: An overview of the trial has been included as Figure 1. A graphical summary can indeed be provided if considered appropriate by the editor.

REFERENCES

Guillot, P. V., C. Götherström, J. Chan, H. Kurata and N. M. Fisk (2007). "Human first-trimester fetal MSC express pluripotency markers and grow faster and have longer telomeres than adult MSC." *Stem Cells* 25(3): 646-654.

Götherström, C., A. West, J. Liden, M. Uzunel, R. Lahesmaa and K. Le Blanc (2005). "Difference in gene expression between human fetal liver and adult bone marrow mesenchymal stem cells." *Haematologica* 90(8): 1017-1026.

Panaroni, C., R. Gioia, A. Lupi, R. Besio, S. A. Goldstein, J. Kreider, S. Leikin, J. C. Vera, E. L. Mertz, E. Perilli, F. Baruffaldi, I. Villa, A. Farina, M. Casasco, G. Cetta, A. Rossi, A. Frattini, J. C. Marini, P. Vezzoni and A. Forlino (2009). "In utero transplantation of adult bone marrow decreases perinatal lethality and rescues the bone phenotype in the knockin murine model for classical, dominant osteogenesis imperfecta." *Blood* 114(2): 459-468.

Yu, Y., A. V. Valderrama, Z. Han, G. Uzan, S. Naserian and E. Oberlin (2021). "Human fetal liver MSCs are more effective than adult bone marrow MSCs for their immunosuppressive, immunomodulatory, and Foxp3(+) T reg induction capacity." *Stem Cell Res Ther* 12(1): 138.

Addition 2024-01-03: After a second request from the Editorial Office of BMJOpen:

New versions of the manuscript (clean and with tracked changes) have been uploaded to the portal after implementing three requested changes (changed reference format, removal of an embedded figure, detailed contributorship statement).

VERSION 2 – REVIEW

REVIEWER	Bishop, Nick
----------	--------------

	University of Sheffield
REVIEW RETURNED	30-Jan-2024
GENERAL COMMENTS	Thank you for addressing my comments
REVIEWER	Naserian, Sina INSERM, Inserm U1197
REVIEW RETURNED	29-Jan-2024
GENERAL COMMENTS	The authors have addressed all raised questions and concerns. I have no further comments. I want to emphasize the important help from Dr. Estelle Oberlin from Inserm U1197 to review this article.

VERSION 2 – AUTHOR RESPONSE

Reviewer: 3

Dr. Sina Naserian, INSERM

Comments to the Author:

The authors have addressed all raised questions and concerns.

I have no further comments.

I want to emphasize the important help from Dr. Estelle Oberlin from Inserm U1197 to review this article.

Reviewer: 2

Nick Bishop, University of Sheffield

Comments to the Author:

Thank you for addressing my comments

Reviewer: 3

Competing interests of Reviewer: There is no competing interest to declare.

Reviewer: 2

Competing interests of Reviewer: Global CI for Ultragenyx-sponsored studies of setrusmab in children with OI. Co-CI of Horizon Europe-funded study of losartan in OI